# Influence of the Geometrical Cross-Section Design on the Dynamic Cyclic Fatigue Resistance of NiTi Endodontic Rotary Files—An In Vitro Study

**DOI:** 10.3390/jcm10204713

**Published:** 2021-10-14

**Authors:** Vicente Faus-Llácer, Nirmine Hamoud-Kharrat, María Teresa Marhuenda Ramos, Ignacio Faus-Matoses, Álvaro Zubizarreta-Macho, Celia Ruiz Sánchez, Vicente Faus-Matoses

**Affiliations:** 1Department of Stomatology, Faculty of Medicine and Dentistry, University of Valencia, 46010 Valencia, Spain; fausvj@uv.es (V.F.-L.); nirhak@alumni.uv.es (N.H.-K.); marhuen3@uv.es (M.T.M.R.); ignacio.faus@uv.es (I.F.-M.); celia.ruiz@uv.es (C.R.S.); vicente.faus@uv.es (V.F.-M.); 2Department of Endodontics, Faculty of Health Sciences, Alfonso X el Sabio University, 28691 Madrid, Spain; 3Department of Surgery, Faculty of Medicine and Dentistry, University of Salamanca, 37008 Salamanca, Spain

**Keywords:** endodontics, cyclic fatigue, cross-section design, NiTi, continuous rotation, energy-dispersive X-ray

## Abstract

The aim of this study was to analyze and compare the influence of the geometrical cross-section design on the dynamic cyclic fatigue resistance of NiTi endodontic rotary files. Materials and Methods: Forty sterile endodontic rotary files were selected and distributed into the following study groups: A: 25.06 double S-shaped cross-section NiTi alloy endodontic rotary files (Mtwo) (*n* = 10); B: 20.04 rectangular cross-section NiTi alloy endodontic rotary files (T Pro E1) (*n* = 10); C: 25.04 convex triangular cross-section NiTi alloy endodontic rotary files (T Pro E2) (*n* = 10); and D: 25.06 triangular cross-section NiTi alloy endodontic rotary files (T Pro E4) (*n* = 10). A cyclic fatigue device was used to conduct the static cyclic fatigue tests with stainless steel artificial root canal systems with 200 µm and 250 µm apical diameter, 60° curvature angle, 3 mm radius of curvature, 20 mm length, and 4% and 8% taper. The results were analyzed using the ANOVA test and Weibull statistical analysis. Results: All the pairwise comparisons presented statistically significant differences between the time to failure and number of cycles to failure for the cross-section design study groups (*p* < 0.001). Conclusions: the double S-shaped cross-section of Mtwo NiTi endodontic files shows higher cyclic fatigue resistance than the rectangular cross-section of T Pro E1 NiTi endodontic files, the convex triangular cross-section of T Pro E2 NiTi endodontic files, and the triangular cross-section of T Pro E4 NiTi endodontic files.

## 1. Introduction

The introduction of nickel–titanium alloy (NiTi) in the manufacturing of root canal instruments entailed a great revolution in the field of endodontics, as these endodontic files decreased the iatrogenic complications [1,2]. However, the failure of endodontic rotary files is still a concern, despite the continuous mechanical and chemical improvements in the NiTi alloy endodontic rotary instruments made by manufacturers to reduce the incidence of complications during root canal treatment [3]. Nevertheless, the incidence of fracture of NiTi endodontic rotary files ranges from 0.09% to 5% [4,5]. The failure of NiTi endodontic rotary files occurs when fatigue resistance is overcome by torsional stress, flexural bending (cyclic) stress, or a combination of the two [6]. Specifically, torsional fatigue occurs when the tip of the endodontic file becomes blocked in the root canal while the instrument continues rotating [7], and flexural bending fatigue occurs by the alternating application of compressive and tensile stress cycles on a curved root canal, leading to overcoming plastic deformation and the subsequent failure of the endodontic rotary instrument [6,8].

In addition, the unexpected failure of the NiTi alloy endodontic rotary instruments might condition the outcome of the root canal treatment by blocking the advancement of disinfecting agents beyond the fractured instrument [9,10,11], which may lead to subsequent pulp necrosis and the formation of periapical lesions [12] or decrease the success rate of root canal treatment of teeth with periapical pathology [13]. Therefore, several reports have been conducted to analyze the influence of both the NiTi alloy and the geometrical parameters on the torsional and flexural bending resistance of endodontic rotary instruments to prevent the incidence of failure of endodontic rotary instruments. Both the chemical composition and crystalline structure of the NiTi alloy have been widely found to highly influence the fatigue resistance of endodontic rotary files, in particular, the endodontic rotary systems, composed of a higher concentration of the martensitic phase and manufactured by electropolishing, ion implantation, cryogenic treatment, and heat treatments, improve the mechanical behavior of NiTi endodontic rotary files, increasing their cyclic fatigue resistance [14]. However, some geometrical factors have also been reported to influence the instrument’s performance, including the taper and apical diameter [15], cross-section design [16,17], flute length, helix angle, and pitch [18]. Unfortunately, the independent assessment of each factor associated with flexural bending fatigue may be difficult in a clinical setting due to the heterogeneous anatomy of the root canal system; thus, controlled experimental studies have been conducted to independently analyze each variable using custom-made cyclic fatigue devices [15].

The aim of this study was to analyze and compare the influence of the geometrical cross-section design on the dynamic cyclic fatigue resistance of NiTi endodontic rotary files, with a null hypothesis (H_0_) stating that the geometry of the cross-section design would not affect the resistance of NiTi endodontic rotary files to dynamic cyclic fatigue.

## 2. Materials and Methods

### 2.1. Study Design

Forty (40) sterile and non-used NiTi alloy endodontic rotary instruments were used in this in vitro study. A controlled experimental trial was performed at the Department of Stomatology of the Faculty of Medicine and Dentistry at the University of Valencia (Valencia, Spain), between March and July 2021. The NiTi endodontic rotary files were selected and categorized into the following study groups: A: double S-shaped cross-section with 250 µm apical diameter and 6% taper conventional NiTi alloy endodontic rotary files mainly consisting of austenite phase at body temperature [19] (Ref.: 0236 025 025, Mtwo, VDW, Munich, Germany) (*n* = 10) (Mtwo); B: rectangular cross-section with 200 µm apical diameter and 4% taper austenite phase NiTi alloy endodontic rotary files (Ref.: 20010103, T Pro E1, Perfect Endo, Shenzhen Perfect Medical Instruments, Shanwei City, China) (*n* = 10) (T Pro E1); C: convex triangular cross-section with 250 µm apical diameter and 4% taper austenite phase NiTi alloy endodontic rotary file (Ref.: 20010103, T Pro E2, Perfect Endo, Shenzhen Perfect Medical Instruments, Shanwei City, China) (*n* = 10) (T Pro E2); and D: triangular cross-section with 250 µm apical diameter and 6% taper austenite phase NiTi alloy endodontic rotary file (Ref.: 20010103, T Pro E4, Perfect Endo, Shenzhen Perfect Medical Instruments, Shanwei City, China) (*n* = 10) (T Pro E4). All endodontic rotary files were manufactured in austenitic phase with an austenite finish (*A*_f_), and the temperatures of the Mtwo, T Pro E1, T Pro E2, and T Pro E4 were approximately 15 °C [19], 15 °C, 20 °C, and 20 °C, respectively. The *A*_f_ temperatures of T Pro E1, T Pro E2, and T Pro E4 were provided by the manufacturer.

### 2.2. Scanning Electron Microscopy Analysis

All NiTi endodontic rotary files were initially analyzed under scanning electron microscopy (SEM) (HITACHI S-4800, Fukuoka, Japan) at ×30 and ×600 in the Central Support Service for Experimental Research of the University of Valencia (Burjassot, Spain) under the following exposure parameters: acceleration voltage: 20 kV, magnification from 100× to 6500×, and a resolution between −1.0 nm at 15 kV and 2.0 nm at 1 kV, to perform a surface characterization to discard further surface defects in its manufacture and analyze and compare the geometrical design of the NiTi endodontic rotary files (Figure 1).

### 2.3. Energy-Dispersive X-ray Spectroscopy Analysis

Additionally, an energy-dispersive X-ray spectroscopy (EDX) analysis was performed on all NiTi endodontic rotary files in the Central Support Service for Experimental Research of the University of Valencia (Burjassot, Spain) under the following exposure parameters: acceleration voltage: 20 kV; magnification: from 100× to 6500×; and a resolution between −1.0 nm at 15 kV and 2.0 nm at 1 kV, in order to analyze the elemental composition of the chemical elements of the NiTi endodontic rotary files used in the static fatigue tests, by means of the atomic weight percent measurement, at three randomized locations (Figure 2).

### 2.4. Experimental Cyclic Fatigue Model

Dynamic cyclic fatigue tests were performed using the previously described custom-made device (utility model patent number ES1219520) [20]. The structure of the dynamic cyclic fatigue test device was designed by computer aided design/computer aided engineering (CAD/CAE) 2D/3D software (Midas FX+^®^, Brunleys, Milton Keynes, UK) and created using 3D printing (ProJet^®^ 6000 3D Systems^©^, Rock Hill, SC, USA) (Figure 3).

The custom-made artificial root canals were performed with a 60° curvature according to Schneider’s measuring technique [21] and 3 mm radius of curvature using CAD/CAE 2D/3D software for inverse engineering technology. The artificial root canal was created from stainless steel using electrical discharge machining (EDM) molybdenum wire-cut technology (Cocchiola S.A., Buenos Aires, Argentina). This process ensured intimate contact between the NiTi endodontic reciprocating files and the artificial root canal walls. The artificial root canal was positioned on its support, and failure of the endodontic rotary instrument was detected using a Light-Dependent Resistor (LDR) sensor (Ref.: C000025, Arduino LLC^®^, Ivrea, Italy) located at the apex of the artificial root canal. The LDR sensor quantifies the continuous light source emitted by a high-brightness white Light-Emitting Diode (LED) (20000 mcd) (Ref.: 12.675/5/b/c/20k, Batuled, Coslada, Spain), which is located opposite the artificial root canal. The light signals emitted by the LED sensor were detected by the LDR (Ref.: C000025, Arduino LLC^®^) sensor with a frequency of 50 ms to accurately identify the precise time of failure.

The direction and speed of the movement generated by the brushed DC gear motor (Ref.: 1589, Pololu^®^ Corporation, Las Vegas, NV, USA) and controlled by the driver (Ref.: DRV8835, Pololu^®^ Corporation, Las Vegas, NV, USA) were transferred to the artificial root canal support through a roller bearing system (Ref.: MR104ZZ, FAG, Schaeffler Herzogenaurach, Germany). The artificial root canal support moved in a pure axial motion using a lineal guide (Ref.: HGH35C 10249-1 001 MA, HIWIN Technologies Corp. Taichung, Taiwan). All the NiTi endodontic rotary files were used with a 6:1 reduction handpiece (X-Smart plus, Dentsply Maillefer, Baillagues, Switzerland) and torque-controlled motor. Mtwo NiTi alloy endodontic rotary files (Ref.: 0236 025 025, Mtwo, VDW, Munich, Germany) were used at 250 rpm and 2.3 N/cm torque, T Pro E1 austenite phase NiTi alloy endodontic rotary files (Ref.: 20010103, T Pro E1, Perfect Endo, Shenzhen Perfect Medical Instruments, Shanwei City, China) were used at 250 rpm and 2 N/cm torque, T Pro E2 austenite phase NiTi alloy endodontic rotary files (Ref.: 20010103, T Pro E1, Perfect Endo, Shenzhen Perfect Medical Instruments, Shanwei City, China) were used at 250 rpm and 2 N/cm torque, and T Pro E4 austenite phase NiTi alloy endodontic rotary files (Ref.: 20010103, T Pro E1, Perfect Endo, Shenzhen Perfect Medical Instruments, Shanwei City, China) were used at 250 rpm and 2 N/cm torque, according to the manufacturer’s instructions.

All NiTi endodontic files were used in the dynamic cyclic fatigue device at a frequency of 60 pecking movements/min according to a previous study [20]. To reduce the friction between the rotating files and the artificial canal walls, special high-flow synthetic oil designed for the lubrication of mechanical parts (Singer All-Purpose Oil; Singer Corp., Barcelona, Spain) was applied.

All NiTi endodontic rotary files were used until fracture occurred. The time to failure and the number of cycles to failure were measured and recorded.

### 2.5. Statistical Tests

Statistical analysis of all the variables was carried out using SAS 9.4 (SAS Institute Inc., Cary, NC, USA). Descriptive statistics are expressed as the mean and standard deviation (SD) for quantitative variables. Comparative analysis was performed by comparing the time to failure (in seconds) and the number of cycles to failure using the ANOVA test. For the comparisons, the *p*-values were adjusted using the Tukey method to correct the type I error. In addition, Weibull characteristic strength and Weibull modulus were calculated. The statistical significance was set at *p* ˂ 0.05.

## 3. Results

SEM analysis of the T Pro E2 NiTi endodontic rotary files (Ref.: 20010103, T Pro E1, Perfect Endo, Shenzhen Perfect Medical Instruments, Shanwei City, China) showed accumulation of organic matter, but none of the NiTi endodontic rotary files showed relevant structural alterations. Moreover, manufacturing lines were distributed perpendicularly to the longitudinal axis in all of the endodontic rotary files and also parallel to each other due to the manufacturing process by laser machining. The width and spacing of the manufacturing lines and tubular porosity correspond to the precision and intensity of the laser machining process. In addition, the macroscopically geometrical design of the double S-shaped cross-section of Mtwo NiTi alloy endodontic rotary files (Ref.: 0236025025, Mtwo, VDW, Munich, Germany) showed a higher pitch than the rectangular cross-section of T Pro E1 NiTi endodontic files (Ref.: 20010103, T Pro E1, Perfect Endo, Shenzhen Perfect Medical Instruments, Shanwei City, China), the convex triangular cross-section of T Pro E2 NiTi endodontic files (Ref.: 20010103, T Pro E1, Perfect Endo, Shenzhen Perfect Medical Instruments, Shanwei City, China), and the triangular cross-section of T Pro E4 NiTi endodontic files (Ref.: 20010103, T Pro E1, Perfect Endo, Shenzhen Perfect Medical Instruments, Shanwei City, China).

EDX micro-analysis of the double S-shaped cross-section of Mtwo NiTi alloy endodontic rotary files (Ref.: 0236025025, Mtwo, VDW, Munich, Germany), the rectangular cross-section of T Pro E1 NiTi endodontic files (Ref.: 20010103, T Pro E1, Perfect Endo, Shenzhen Perfect Medical Instruments, Shanwei City, China), the convex triangular cross-section of T Pro E2 NiTi endodontic files (Ref.: 20010103, T Pro E1, Perfect Endo, Shenzhen Perfect Medical Instruments, Shanwei City, China), and the triangular cross-section of T Pro E4 NiTi endodontic files (Ref.: 20010103, T Pro E1, Perfect Endo, Shenzhen Perfect Medical Instruments, Shanwei City, China) was performed at 20 kV as this allowed a deeper analysis of the NiTi endodontic rotary files surface. In summary, the double S-shaped cross-section of Mtwo NiTi alloy endodontic rotary files (Ref.: 0236025025, Mtwo, VDW, Munich, Germany) differs in the chemical elements present in the metal alloy, in accordance with the rectangular cross-section of T Pro E1 NiTi endodontic files (Ref.: 20010103, T Pro E1, Perfect Endo, Shenzhen Perfect Medical Instruments, Shanwei City, China), the convex triangular cross-section of T Pro E2 NiTi endodontic files (Ref.: 20010103, T Pro E1, Perfect Endo, Shenzhen Perfect Medical Instruments, Shanwei City, China), and the triangular cross-section of T Pro E4 NiTi endodontic files (Ref.: 20010103, T Pro E1, Perfect Endo, Shenzhen Perfect Medical Instruments, Shanwei City, China), which include aluminum in the chemical composition of the metal alloy (Table 1).

The mean and SD values for the time to failure (in seconds) for each of the study groups are displayed in Table 2 and Figure 4.

The ANOVA test showed statistically significant differences between the time to failure of all NiTi endodontic rotary files (*p* ˂ 0.001) (Figure 4). The results related to the number of cycles to failure are similar as the dynamic cyclic fatigue device had a frequency of 60 pecking movements/min.

The scale distribution parameter (η) of the Weibull statistical analysis found statistically significant differences between the time to failure of all NiTi endodontic rotary files (*p* ˂ 0.001) (Table 3, Figure 5). However, the shape distribution parameter (β) of the Weibull analysis found no statistically significant differences between the time to failure of any of the NiTi endodontic rotary files (*p* ˃ 0.05). The results related to the number of cycles to failure are similar as the dynamic cyclic fatigue device had a frequency of 60 pecking movements/min (Table 3, Figure 5).

## 4. Discussion

The results obtained in the present study reject the null hypothesis (H_0_) that stated that the geometry of the cross-section design would not affect the resistance of NiTi endodontic rotary files to dynamic cyclic fatigue.

The results derived in the present study reported that Mtwo NiTi alloy endodontic rotary files with double S-shaped cross-section showed higher resistance to dynamic cyclic fatigue than T Pro E1 austenite phase NiTi alloy endodontic rotary files with rectangular cross-sections, T Pro E2 austenite phase NiTi alloy endodontic rotary files with convex triangular cross-sections, and T Pro E4 austenite phase NiTi alloy endodontic rotary files with triangular cross-sections. The results can be summarized in that with the increase in the mass and the contact points between the instrument surface and the dentin walls of the root canal, the cyclic fatigue resistance of the NiTi endodontic rotary files decreases. This can also influence the flexibility of the NiTi endodontic rotary files and lead the instrument to cause excessive root canal dentine removal, apical transportation [22], root perforations, and fractures [4,23,24].

The persistent bacterial load present in the root canal system after endodontic therapy has been highlighted as a relevant etiologic factor in the endodontic failure and secondary endodontic infections [25]; moreover, Sjögren established a relationship between the bacterial load reduction during the root canal treatment and the prognosis of the endodontic therapy, and reported that negative microbiological cultures obtained from the root canal system led to an endodontic success rate close to 94%, whereas positive cultures reduced the success rate to 68% [26]. This is the reason that the cyclic fatigue resistance of NiTi endodontic rotary files has been widely analyzed.

The design of the anatomical-based artificial root canal used in the present study was based on the method described by Schneider [21], selecting a 5 mm radius and 60° curvature angle and adapting the geometry to the NiTi endodontic rotary files included in this study. Previous studies have shown that the fatigue resistance of endodontic rotary files decreases as the angle of curvature increases and the radius of curvature decreases [10,27,28], since the stress accumulation on the endodontic rotary file is inversely proportional to the radius of curvature of the canal. As a result, in more abrupt root canals, there is an augmentation of the torsion and flexural bending fatigue that ultimately results in instrument fracture [10,21]. Moreover, clinical or even ex vivo experimental studies would be desirable to reproduce clinical conditions and extrapolate the cyclic fatigue results to the clinical setting; however, the difficulty to homogenize the radius, curvature angle, apical diameter, hardness, and cross-section of the root canals can bias the study by introducing more variables [28]. Therefore, custom-made static and dynamic cyclic fatigue devices have been developed to independently analyze the influence of the variable under study; unfortunately, there is neither a norm that regulates the characteristics of the custom-made cyclic fatigue devices nor an international standard for testing the cyclic fatigue behavior of NiTi endodontic rotary instruments with taper higher than 2% [29].

Static and dynamic testing devices have been used to analyze the cyclic fatigue. In the static cyclic fatigue testing models, the NiTi endodontic files are rotated until fracture occurs and the tension–compression cycles are concentrated in the maximum curvature angle of the root canal, resulting in microstructural alterations in the file and subsequent failure. Therefore, dynamic cyclic fatigue testing devices are preferable to better reproduce the clinical conditions, especially the pecking motion of the NiTi endodontic rotary files. Thus, this study used a dynamic cyclic fatigue testing model, an anatomical-based artificial root canal and an automatic detection system to objectively and accurately identify failures of endodontic rotary files [28,30,31].

Previous studies have analyzed the influence of cross-section design on the mechanical behavior of the NiTi endodontic rotary files. Sekar et al. analyzed the role of the cross-section on the cyclic fatigue resistance of NiTi endodontic rotary files under continuous and reciprocation motion and reported that the 25.06 Mtwo rotary files were significantly more resistant to failure than Revo-S SU and One Shape files in both continuous (*p* < 0.001) and reciprocating motion (*p* < 0.001) [17]. These findings are consistent with the results of our study, which concluded that the double S-shaped cross-section of Mtwo NiTi endodontic files showed higher cyclic fatigue resistance than the rectangular cross-section of T Pro E1 NiTi endodontic files, the convex triangular cross-section of T Pro E2 NiTi endodontic files, and the triangular cross-section of T Pro E4 NiTi endodontic files. In addition, de Menezes et al. reported that ProDesign endodontic rotary files with a modified double S-shaped cross-section design presented a significantly higher (*p* < 0.05) number of cycles to failure (910.37 ± 472.10) than Wave One Gold endodontic reciprocating files with a parallelogram cross-section design (264.76 ± 305.42) in artificial root canals with a 60° curvature and 5 mm radius of curvature [32]. Moreover, Adiguzel et al. showed that XP-endo Shaper endodontic rotary files with triangular cross-sections design presented a significantly higher (*p* < 0.05) number of cycles to failure (3064.0 ± 248.1) than HyFlex CM endodontic rotary files with a variable cross-section design (from triangular to trapezoidal and quadratic) (1120.5 ± 106.1) in artificial root canals with a 60° curvature and 3 mm radius of curvature [33]; however, Uygun et al. showed that HyFlex EDM endodontic rotary files with a variable cross-section design (from triangular to trapezoidal and quadratic) presented a significantly higher (*p* < 0.05) number of cycles to failure (1710.42 ± 114.89) than Vortex Blue endodontic rotary files with a convex triangular cross-section design (548.39 ± 77.64), ProTaper Gold endodontic rotary files with a convex triangular cross-section design (600.83 ± 66.49), and One Curve endodontic rotary files with a variable cross-section design (from double S-shaped to triangular) (959.58 ± 61.18) in artificial root canals with a 60° curvature and 3 mm radius of curvature [34].

Unfortunately, the limitations of the present study prevented the analysis of more cross-section designs to standardize the NiTi alloy, apical diameter, pitch, helix angle, manufacturing process, speed, and taper. In addition, the study was not developed in a clinical environment due to the difficulty in standardizing the sample.

## 5. Conclusions

The conclusion derived from the present study is that the double S-shaped cross-section of Mtwo NiTi endodontic files shows higher cyclic fatigue resistance than the rectangular cross-section of T Pro E1 NiTi endodontic files, the convex triangular cross-section of T Pro E2 NiTi endodontic files, and the triangular cross-section of T Pro E4 NiTi endodontic files.

## Figures and Tables

**Figure 1 jcm-10-04713-f001:**
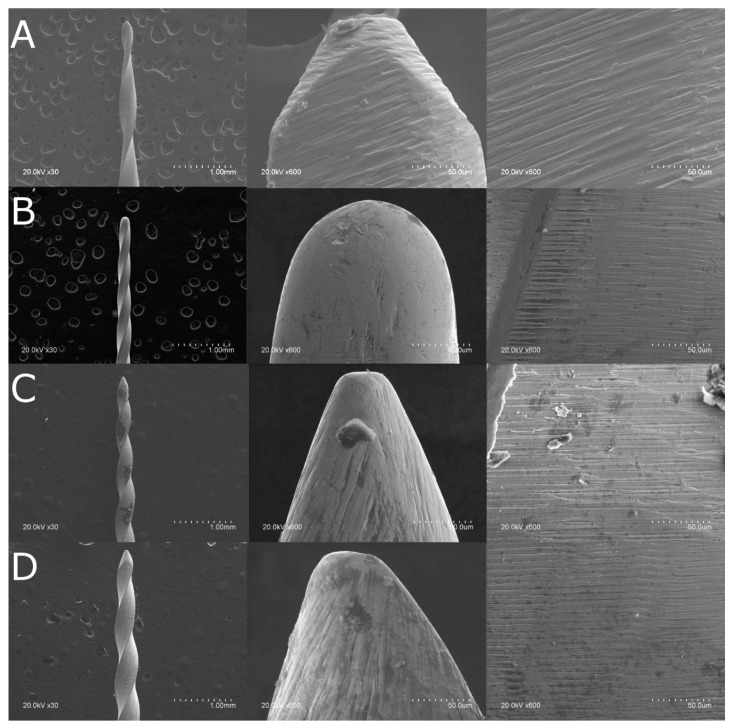
(**A**) SEM analysis of the Mtwo NiTi alloy endodontic rotary file, (**B**) T Pro E1 Gold-Wire NiTi alloy endodontic rotary file, (**C**) T Pro E2 Gold-Wire NiTi alloy endodontic rotary file, and (**D**) T Pro E4 Gold-Wire NiTi alloy endodontic rotary file.

**Figure 2 jcm-10-04713-f002:**
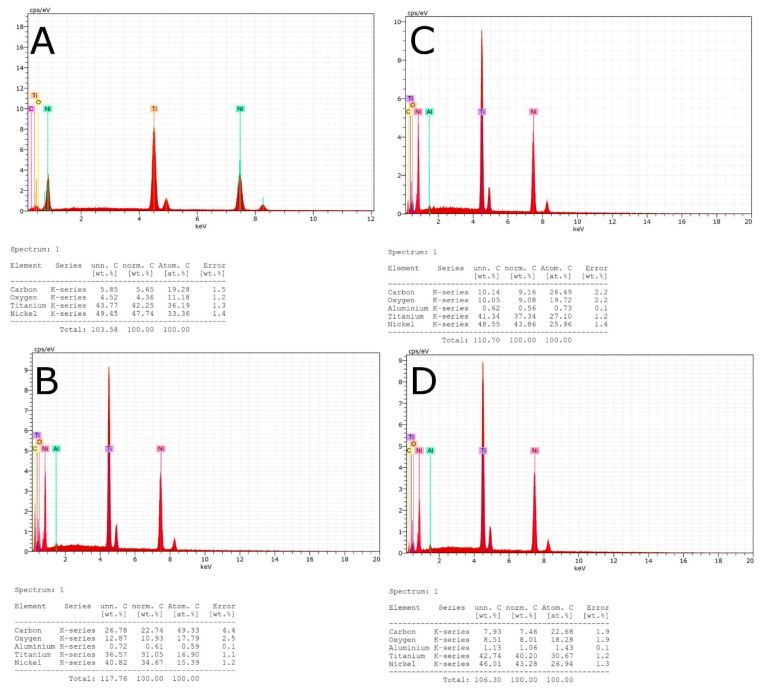
(**A**) EDX micro-analysis of the Mtwo NiTi alloy endodontic rotary file, (**B**) T Pro E1 austenite phase NiTi alloy endodontic rotary file, (**C**) T Pro E2 austenite phase NiTi alloy endodontic rotary file, and (**D**) T Pro E4 austenite phase NiTi alloy endodontic rotary file.

**Figure 3 jcm-10-04713-f003:**
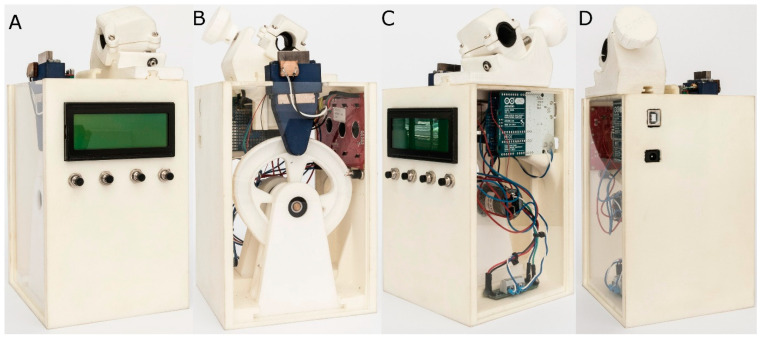
(**A**) Front, (**B**) back, (**C**) right, and (**D**) left surfaces of the dynamic cyclic fatigue device.

**Figure 4 jcm-10-04713-f004:**
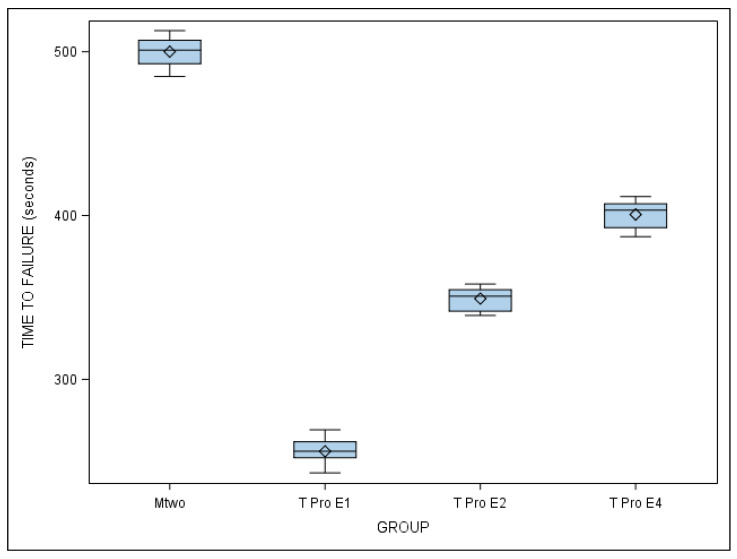
Box plot of the time to failure of Mtwo NiTi alloy endodontic rotary files, T Pro E1 austenite phase NiTi alloy endodontic rotary files, T Pro E2 austenite phase NiTi alloy endodontic rotary files, and T Pro E4 austenite phase NiTi alloy endodontic rotary files.

**Figure 5 jcm-10-04713-f005:**
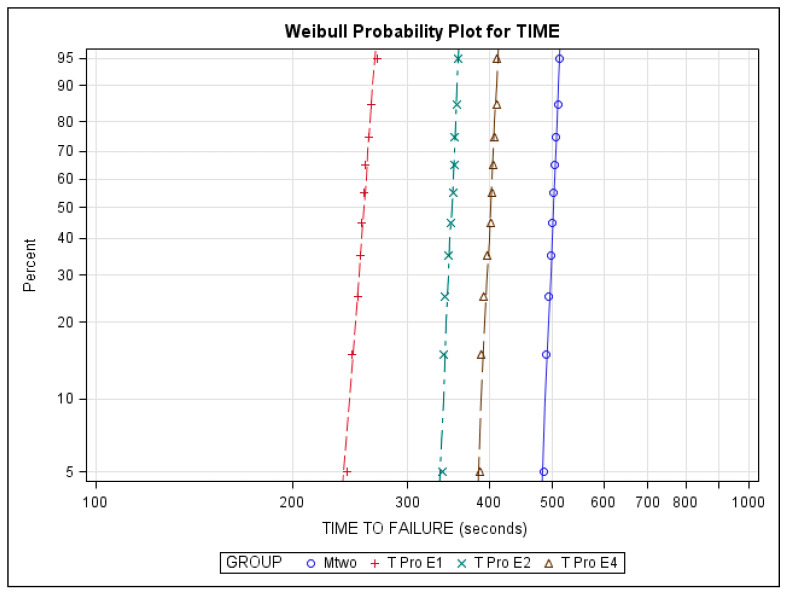
Weibull probability plot of time to failure of Mtwo NiTi alloy endodontic rotary files, T Pro E1 austenite phase NiTi alloy endodontic rotary files, T Pro E2 austenite phase NiTi alloy endodontic rotary files, and T Pro E4 austenite phase NiTi alloy endodontic rotary files.

**Table 1 jcm-10-04713-t001:** Mean atomic weight percent (%) of Mtwo NiTi alloy endodontic rotary files, T Pro E1 austenite phase NiTi alloy endodontic rotary files, T Pro E2 austenite phase NiTi alloy endodontic rotary files, and T Pro E4 austenite phase NiTi alloy endodontic rotary files.

Spectrum	C	O	Al	Ti	Ni
Mtwo 20 kV (1–3)	20.92	10.89	-	37.60	30.58
T Pro E1 20 kV (1–3)	42.51	21.15	0.52	19.48	16.34
T Pro E2 20 kV (1–3)	26.49	19.72	0.73	27.10	25.96
T Pro E4 20 kV (1–3)	39.52	19.25	2.43	20.58	18.23

**Table 2 jcm-10-04713-t002:** Descriptive statistics of the time to failure of Mtwo NiTi alloy endodontic rotary files, T Pro E1 austenite phase NiTi alloy endodontic rotary files, T Pro E2 austenite phase NiTi alloy endodontic rotary files, and T Pro E4 austenite phase NiTi alloy endodontic rotary files.

Study Group	*n*	Mean	SD	Minimum	Maximum
Mtwo	10	500.06 ^a^	9.22	484.90	512.90
T Pro E1	10	256.05 ^b^	7.96	242.90	269.20
T Pro E2	10	349.29 ^c^	7.02	339.00	358.20
T Pro E14	10	400.64 ^d^	8.72	387.10	411.60

^a,b,c,d^ Statistically significant differences between groups (*p* < 0.05).

**Table 3 jcm-10-04713-t003:** Weibull statistics of time to failure of replica-like and original brand NiTi endodontic rotary files study groups.

	Weibull Shape (β)				Weibull Scale (η)			
Study Group	Estimate	St Error	Lower	Upper	Estimate	St Error	Lower	Upper
Mtwo	67.0256	16.7296	41.0943	109.3202	504.2430	2.5132	499.3413	509.1928
T Pro E1	37.0114	8.8527	23.1599	59.1472	259.6936	2.3498	255.1287	264.3402
T Pro E2	64.0224	16.4767	38.6606	106.0220	352.4523	1.8358	348.8725	356.0689
T Pro E14	59.2617	15.1900	35.8586	97.9388	404.5474	2.2760	400.1110	409.0329

## Data Availability

Data are available on request due to restrictions, e.g., privacy or ethical.

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
