# Peer review of "Influence of the Geometrical Cross-Section Design on the Dynamic Cyclic Fatigue Resistance of NiTi Endodontic Rotary Files—An In Vitro Study"

_jcm, 2021, doi:10.3390/jcm10204713_

Round 1

Reviewer 1 Report

Dear Authors,

  1. In your study all the files are in the austenitic phase. Please write this information in case of Mtwo files. If it will be possible to find, please add the Af (austenite finish) temperature to confirm that all the files are in the austenitic phase.
  2. There is a discrepancy between radius of curvature, once in (123) there is 3 mm radius of curvature when in Abstract is 5 mm radius of curvature
  3. (127) … NiTi endodontic reciprocating files and the artificial root canal. Please re-write between the working NiTi endodontic files.
  4. (156) To reduce the friction between the reciprocating files and the artificial canal walls In my opinion, is better to write to reduce the friction between the rotating files and the artificial…./ Reciprocation is one of the movements and this file not working under this condition. During your study you use rotation with parameters of torque and rpm.  
  1. (249) apical transport [ xxi]à apical transportation
  2. (288) MTwo- should be Mtwo as in the whole text
  3. In the discussion part:
  • There should be more studies to compare with as you mentioned in the sentence: Previous studies have analyzed the influence of cross-section design on the mechanical behavior of the NiTi endodontic rotary files. You discuss only with one study (Sekar et al). Please find more research with the same condition (dynamic model, 60-degree, 5 mm curvature, and continuous movement).
  • Eymirli A, Uzunoğlu Özyürek E, Serper A. Sealer penetration: effect of separated file's cross-section, taper and motion characteristics. Clin Oral Investig. 2021, 25, 1077-1084 this study is not connected with your research
  • You write about articles which in the methodology used the finite element methods, but as I have mentioned, it is not the same condition as in your study thus these articles should not be listed to discuss with (Xu et al, Kim et al.)

Author Response

Dear Reviewer 1:

I’m pleased to resubmit the manuscript of the work entitled, “Influence of the Geometrical Cross-section Design on the Dynamic Cyclic Fatigue Resistance of NiTi Endodontic Rotary Files. An In Vitro Study”

Reviewer 1: I don't feel qualified to judge about the English language and style

Response: In order to adapt to the reviewer's 1 comments, we have send the manuscript to the English Editing Service of MDPI. We attached the Certificate.

Reviewer 1: In your study all the files are in the austenitic phase. Please write this information in case of Mtwo files. 

Response: In order to adapt to the reviewer's 1 comments, we have added this information in the Material and Methods section.

Reviewer 1:  If it will be possible to find, please add the Af (austenite finish) temperature to confirm that all the files are in the austenitic phase.

Response: In order to adapt to the reviewer's 1 comments, we clarify that all endodontic rotary files were manufactured in austenitic phase with an autenite finish (Af) temperature of the Mtwo, T Pro E1, T Pro E2 and T Pro E4 of approximately 15°C, 15°C, 20°C and 20°C, respectively. The Af temperatures of T Pro E1, T Pro E2 and T Pro E4 were provided by the manufacturer.

Reviewer 1: There is a discrepancy between radius of curvature, once in (123) there is 3 mm radius of curvature when in Abstract is 5 mm radius of curvature

Response: In order to adapt to the reviewer's 1 comments, we have corrected the radius of curvature of the Abstract section

Reviewer 1: (156) To reduce the friction between the reciprocating files and the artificial canal walls In my opinion, is better to write to reduce the friction between the rotating files and the artificial…./ Reciprocation is one of the movements and this file not working under this condition. During your study you use rotation with parameters of torque and rpm.

Response: In order to adapt to the reviewer's 1 comments, we have changed the word.

Reviewer 1: (249) apical transport [ xxi]à apical transportation

Response: In order to adapt to the reviewer's 1 comments, we have changed the word.

Reviewer 1: (288) MTwo- should be Mtwo as in the whole text

Response: In order to adapt to the reviewer's 1 comments, we have changed the word.

Reviewer 1: There should be more studies to compare with as you mentioned in the sentence: Previous studies have analyzed the influence of cross-section design on the mechanical behavior of the NiTi endodontic rotary files. You discuss only with one study (Sekar et al). Please find more research with the same condition (dynamic model, 60-degree, 5 mm curvature, and continuous movement).

Response: In order to adapt to the reviewer's 1 comments, we have added more references related to the present study at the Discussion section.

Reviewer 1: Eymirli A, Uzunoğlu Özyürek E, Serper A. Sealer penetration: effect of separated file's cross-section, taper and motion characteristics. Clin Oral Investig. 2021, 25, 1077-1084 this study is not connected with your research

Response: In order to adapt to the reviewer's 1 comments, we have removed the sentence and the reference.

Reviewer 1: You write about articles which in the methodology used the finite element methods, but as I have mentioned, it is not the same condition as in your study thus these articles should not be listed to discuss with (Xu et al, Kim et al.)

Response: In order to adapt to the reviewer's 1 comments, we have removed the sentences and articles related to finite element methods.

We take this opportunity to thank the recommendations and suggestions made by the reviewers to improve the document.

Yours sincerely,

Reviewer 2 Report

This manuscript is well written, the methods used are standard, and the influence of the geometrical cross-section design of endodontic files on their cyclic fatigue resistance is demonstrated. That being said,  the information on metallurgy and manufacturing of the files (e.g: made in Germany or in China) is not considered and thus the influence of design can not be separated from influnces of metallurgy, heat treatment and other variables. Thus the experiment results can not support the conclusions. 

Author Response

Dear Reviewer 2:

I’m pleased to resubmit the manuscript of the work entitled, “Influence of the Geometrical Cross-section Design on the Dynamic Cyclic Fatigue Resistance of NiTi Endodontic Rotary Files. An In Vitro Study”

Reviewer 2: English language and style are fine/minor spell check required

Response: In order to adapt to the reviewer's 2 comments, we have send the manuscript to the English Editing Service of MDPI. We attached the Certificate.

Reviewer 2: That being said,  the information on metallurgy and manufacturing of the files (e.g: made in Germany or in China) is not considered and thus the influence of design can not be separated from influnces of metallurgy, heat treatment and other variables. Thus the experiment results can not support the conclusions.

Response: In order to adapt to the reviewer's 2 comments, we clarify that each instrumentation system is unique and is made up of different geometric (pitch, flute, taper, tip, helix angle, cross section), metallurgical (chemical composition, atomic weight, heat treatment, manufacturing process) and kinetic (type of movement, speed) characteristics, which make it difficult to compare the different systems. For this reason, an attempt has been made to select files that are the most similar to each other. This fact has been highlighted as a limitation of the study in the Discussion section.

We take this opportunity to thank the recommendations and suggestions made by the reviewers to improve the document.

Yours sincerely,

This manuscript is a resubmission of an earlier submission. The following is a list of the peer review reports and author responses from that submission.